# Topological Refraction in Kagome Split-Ring Photonic Insulators

**DOI:** 10.3390/nano12091493

**Published:** 2022-04-28

**Authors:** Huichang Li, Chen Luo, Tailin Zhang, Jianwei Xu, Xiang Zhou, Yun Shen, Xiaohua Deng

**Affiliations:** 1School of Physics and Materials Science, Nanchang University, Nanchang 330031, China; lhc_qx@163.com (H.L.); nc_lc@email.ncu.edu.cn (C.L.); 400603420025@email.ncu.edu.cn (T.Z.); xjw799576510@163.com (J.X.); zx610080@163.com (X.Z.); 2Institute of Space Science and Technology, Nanchang University, Nanchang 330031, China

**Keywords:** topological phase transition, valley-Hall-like topology, edge states, topological refraction

## Abstract

A valley-Hall-like photonic insulator based on $C_{3v}$ Kagome split-ring is proposed. Theoretical analysis and numerical calculations illustrate that $C_{3v}$ symmetry can be broken not only by global rotation α but also individual rotation θ of the split rings, providing topological phase transitions. Furthermore, refraction of the edge state from the interface into the background space at Zigzag termination is explored. It is shown that positive/negative refraction of the outgoing beam depends on the type of valley (*K* or $K^{\prime }$), from which the edge state is projected. These results provide a new way to manipulate terahertz wave propagation and facilitate the potential applications in directional collimation, beam splitting, negative refraction image, etc.

## 1. Introduction

Photonic topological insulators support some of the most fascinating properties for signal transport and show excellent potential applications in modern optical devices [1,2,3,4,5,6,7], such as reflectionless waveguides, topological quantum interfaces, splitters, robust delay lines, etc.

The key to topological phase transition lies in opening an energy gap in the band structure at certain degenerate points by breaking either the time-reversal symmetry (TRS) or inversion symmetry, which can be realized in classical wave systems mimicking the quantum Hall effect (QHE) [8,9,10,11,12,13,14], the quantum spin Hall effect (QSHE) [15,16,17,18,19,20,21,22,23], or the quantum valley Hall effect (QVHE) [24,25,26,27,28,29,30,31,32,33]. In analogues of the QHE system, breaking of TRS can be achieved by applying a magnetic field or an effective ‘magnetic’ field from circulating fluid flows [10,11,12] or gyroscopes [13,14]. However, this strategy remains challenging due to the structural complexity. In mimicking QSHE or QVHE systems, breaking the symmetry does not require any external fields. Generally, analogues of QSHE can be provided by constructing the system to have two pseudospins, either integrating the different wave polarization degree of freedom or folding the band structure to form a double Dirac cone [15,16,17,18,19,20,21,22,23,30]. For emulating QVHE, such as $C_{3v}$ symmetry photonic crystals composed of triangular rods [6,24,34], square lattices containing square inclusions [35], the new class of valley-Hall-like topological insulators can be designed by breaking the inversion symmetry of the systems [36,37,38].

So far, varieties of topological insulators have been demonstrated. However, most of the current studies focus on the robust wave propagation along the topological interfaces [10,11,12,13,14,31,32,39,40,41]. Further exploration for applicable topological devices, including outcoupling effects or wave manipulations, need to be taken into consideration. More recently, topological positive/negative, near-zero refraction and wave splitting have been reported in valley crystal [25,29,30,31], Weyl [42], and QSHE inspired systems [43,44], which may have potential applications in directional collimation [45], beam splitting [46,47,48], and others.

In this work, a new kind of valley-Hall-like photonic insulator based on two-dimensional (2D) $C_{3v}$ Kagome split-ring is proposed. Theoretical analysis and numerical calculations by COMSOL Multiphysics illustrate that $C_{3v}$ symmetry can be broken not only by global rotation α but also by individual rotation θ of the split rings, providing topological phase transitions. Further, topological positive/negative refraction of the edge states from the interface into background space at Zigzag termination is explored. The results show that refraction of the outgoing beam depends on the type of valley (*K* or $K^{\prime }$) from which the edge state is projected.

## 2. Model and Calculation Method

### 2.1. Model

The 2D unit cell of $C_{3v}$ Kagome split-ring photonic insulator (SRPI) is shown in Figure 1a, where the lattice constant, and the inner and outer diameters of each split ring are *a*, $r_{1}$, and $r_{2}$, respectively. The background material is silicon (Si) with permittivity ε=11.7, and the split ring is set as a perfect electric conductor. The proposed $C_{3v}$ symmetry-broken unit cell to provide non-trivial valley-Hall-like topological states is demonstrated in Figure 1b, in which α and θ are the global and individual rotation of the split rings. As a=400
µm, $r_{1}=0.1a$, $r_{2}=0.16a$, the photonic bands (TM modes with magnetic field along *z* axis) for $\alpha =0^{\circ }$, $\theta =0^{\circ }$, and $\alpha =30^{\circ }$, $\theta =0^{\circ }$ are shown in Figure 1c with blue and red curves, respectively. The existing bandgap closes for $\alpha =0^{\circ }$ and opens for $\alpha =30^{\circ }$.

### 2.2. Valley Chern Number

In SRPI, the Dirac cone near the *K* point can be approximately regarded as a two-band model, and its effective Hamiltonian has the following form through the k·p perturbation method [24,49,50]:(1)$$H_{K}\left(\delta k\right)=\upsilon_{D}\left(\delta k_{x}\sigma_{x}+\delta k_{y}\sigma_{y}\right)+m\upsilon_{D}^{2}\sigma_{z}$$
where $\upsilon_{D}$ is the group velocity at the degenerate Dirac cone, δk represents the momentum deviation from the *K* point, σ is Pauli matrix, $m=\pi \left(f_{1}-f_{2}\right)/\upsilon_{D}^{2}$, $f_{1}$ and $f_{2}$ separately represent the two frequencies at the boundaries of bandgap after the Dirac point lifted. The Berry curvature can be obtained through effective Hamiltonian [49,51]:(2)$$\Omega \left(\delta k\right)=\frac{m\upsilon_{D}}{2{\left(\delta k^{2}+m^{2}\upsilon_{D}^{2}\right)}^{\frac{3}{2}}}$$

The valley Chern number can be obtained by the numerical integration of Berry curvature around the *K* valley:(3)$$C_{K}=\frac{1}{2\pi }\int \Omega \left(\delta k\right)dk^{2}=\frac{sgn[m]}{2}=\frac{sgn(f_{1}-f_{2})}{2}$$

Equation (Equation 3) shows that the valley Chern number only depends on the sign of the Dirac mass *m* ($C_{K}=\frac{1}{2}$ for m>0 and $C_{K}=-\frac{1}{2}$ for m<0). The band inversion means the reversal of $f_{1}$ and $f_{2}$, and consequently the signs of valley Chern number $C_{K}$.

## 3. Results

### 3.1. Band Inversion of Topological Valley-Hall-like States

Typically, the process of band inversion can make the bandgap open then close and then reopen, resulting in the topological phase transition at the *K* (or $K^{\prime }$) point [52]. For our proposed Kagome split-ring photonic insulator shown in Figure 1b, the topological phase transition depending on both global α and individual θ rotation is illustrated in Appendix A. Specifically, for $\theta =0^{\circ }$, the topological phase varying with global α is shown in Figure 2a, in which the original $C_{3v}$ symmetry at $\alpha =0^{\circ }$ is reduced to the $C_{3}$ symmetry. Accordingly, the twofold Dirac degeneracy at the *K* (or $K^{\prime }$) point is lifted. The field distributions of the two eigenstates at the *K* point (denoted as *p* and *q*) for $\alpha =-30^{\circ }$ and $30^{\circ }$, respectively, are illustrated in Figure 2b. When the degeneracy is lifted, the frequency order of the *p* and *q* states flips, indicating a typical band inversion and topological phase transition. According to Equation (Equation 3), the signs of Dirac mass *m* ($sgn\left[m\right]=sgn\left[f_{p}-f_{q}\right]$) for bandgap *I* and II in Figure 2a are opposite, and the difference of valley Chern number *I* and *II* is $\left|\Delta C|=\right|C_{K}^{I}-C_{K}^{II}\left|=\right|\frac{1}{2}-\left(-\frac{1}{2}\right)|=1$, which implies that valley-Hall-like topological edge states can exist at the interface of system composed of SRPIs with distinct valley-Hall-like phases (Section 3.2).

As $\alpha =30^{\circ }$, the topological phase varying with global θ is shown in Figure 2c. The field profiles of the eigenstates at the *K* point for $\theta =-180^{\circ }$, $-110^{\circ }$, $-70^{\circ }$, and $0^{\circ }$ are shown in Figure 2d. Similar to Figure 2b, the frequency order of the eigenstates flips for different θ. With two different bands in Figure 2b involved in the topological phase transition, three bands are involved in Figure 2c, where two different kinds of band inversions (one is between III (III) and IV (VI), and the other is between IV (VI) and *V* (*V*)) exist. In regions III (III) and IV (VI), the signs of Dirac mass *m* ($sgn\left[m\right]=sgn\left[f_{s}-f_{q}\right]$) are opposite, and the difference of their valley Chern numbers is $\left|\Delta C|=\right|C_{K}^{III}-C_{K}^{IV}|=|-\frac{1}{2}-\frac{1}{2}|=1$, which guarantees the existence of topological edge states (Appendix B). Similarly, for IV (VI) and *V* (*V*), opposite signs of Dirac mass *m* ($sgn\left[m\right]=sgn\left[f_{p}-f_{q}\right]$) and different valley Chern numbers ($\left|\Delta C|=\right|C_{K}^{IV}-C_{K}^{V}\left|=\right|\frac{1}{2}-\left(-\frac{1}{2}\right)|=1$) are obtained, providing topological edge states (Appendix B).

### 3.2. Valley Topological Refraction

It is known that the direction of the outgoing beam of the topological edge state depends on the type of valley (*K* or $K^{\prime }$), from which the edge state is projected [45]. Hence, a wave launched from left to right along positive-type and negative-type interfaces will be projected from the *K* and $K^{\prime }$ valley, respectively. To illustrate the refraction of the radiated beam, a topological insulator in Figure 3a is designed, and refractions of edge states from the interface into the background space at Zigzag termination are explored.

Figure 3b shows the dispersion relation of the topological insulator containing different interfaces. Dashed black curves represent the bulk modes, and the dashed red/blue curves represent the negative-type/positive-type interfaces edge states. For different points at the interfaces edge states, the distributions of fields and Poynting vectors (represented by arrows) are shown in Figure 3c.

The *k*-space analysis on out-coupling of *K* projected edge states along the positive-type (Zigzag) interface is demonstrated in Figure 4a. The white solid hexagon represents the first Brillouin zone, and red solid circles show the dispersion in background material Si, in which the incident wavevector *K* will be matched with the equifrequency curve of Si to determine the propagation direction of the radiated beam. The simulated distribution of fields at frequency f=0.102 THz is illustrated in the bottom panel. For out-coupling of $K^{\prime }$ projected edge state along the negative-type (Zigzag) interface, the corresponding analyses are shown in Figure 4b. Obviously, the direction of the outgoing beam in Figure 4a is different with that in Figure 4b due to the different type of valley (*K* or $K^{\prime }$) from which the edge state is projected. Further, the theoretical refraction angle can be quantitatively determined by the phase-matching condition $k\cdot e_{term}=K\cdot e_{term}$ at the terminal (parallel to $e_{term}$), which is $\left|k\right|\cdot \cos\left(60^{\circ }-\theta_{rK}\right)=\left|K\right|\cdot \cos60^{\circ }$ and $\left|k\right|\cdot \cos\left(120^{\circ }+\theta_{rK^{\prime }}\right)=\left|K\right|\cdot \cos60^{\circ }$, for positive-type and negative-type interfaces, respectively. Here, the equifrequency curves *k* of background Si can be determined by $\left|k\right|=2\pi \cdot \frac{f\cdot n_{Si}}{c}$, where *f* represents the incident frequency, *c* is light speed in air, and $n_{Si}=3.42$ is the refractive index of Si. Consequently, $\theta_{rK}=15.78^{\circ }$ and $\theta_{rK^{\prime }}=-75.78^{\circ }$ are obtained in Figure 4a,b. We note that the topological refraction and transmission in our SRPI can maintain strong robustness (Appendix C).

## 4. Conclusions

A valley-Hall-like photonic insulator based on $C_{3v}$ Kagome split-ring is proposed. Theoretical analysis and numerical calculations illustrate that $C_{3v}$ symmetry can be broken by either the global rotation α or individual rotation θ of split rings, providing topological phase transitions. Additionally, refractions of the outgoing beam from the interface into the background space at Zigzag termination are explored. We show that positive/negative refraction can be obtained, which is determined by the type of valley (*K* or $K^{\prime }$). These results provide a new way to manipulate THz wave propagation and facilitate the potential applications in directional collimation, beam splitting, and negative refraction image.

## Figures and Tables

**Figure 1 nanomaterials-12-01493-f001:**
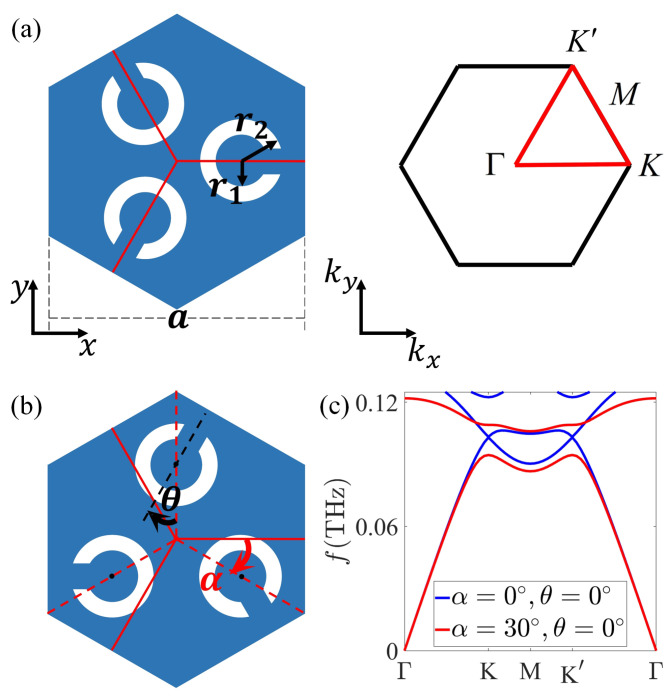
(**a**) The 2D unit cell and first Brillouin zone of $C_{3v}$ Kagome SRPI. The geometric parameters are taken as a=400 µm, $r_{1}=0.1a$, $r_{2}=0.16a$. (**b**) $C_{3v}$ symmetry-broken unit cell, α and θ are the global and individual rotation of the split rings. (**c**) Photonic bands of the SRPI for $\alpha =0^{\circ }$, $\theta =0^{\circ }$ (blue solid curves) and $\alpha =30^{\circ }$, $\theta =0^{\circ }$ (red solid curves).

**Figure 2 nanomaterials-12-01493-f002:**
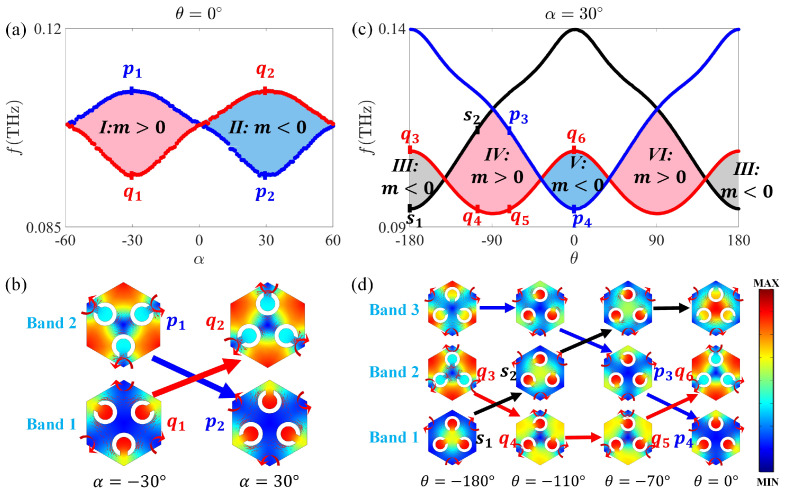
(**a**) Dependence of topological phase transition on global α as $\theta =0^{\circ }$ at the *K* point; the region *I* and II represent bandgaps with m>0 and m<0, respectively. (**b**) The field distribution of the two eigenstates at the *K* point (denoted as *p* and *q*) for $\alpha =-30^{\circ }$ and $30^{\circ }$. (**c**) Dependence of topological phase transition on individual θ as $\alpha =30^{\circ }$ at the *K* point; the region III, IV (VI) and *V* represent bandgaps with m<0, m>0, and m<0, respectively. (**d**) The field distribution of three eigenstates at the *K* point (denoted as *s*, *p*, and *q*) for $\theta =-180^{\circ }$, $-110^{\circ }$, $-70^{\circ }$, and $0^{\circ }$. The Poynting vectors are represented by red arrows.

**Figure 3 nanomaterials-12-01493-f003:**
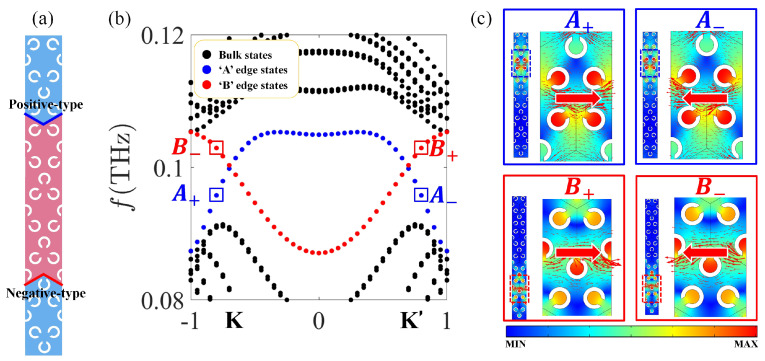
(**a**) Supercell and (**b**) bands of topological insulator with $\alpha =-30^{\circ },\theta =0^{\circ }$ (light red) and $\alpha =30^{\circ },\theta =0^{\circ }$ (light blue). In (**b**), dashed black curves represent the bulk modes and the dashed red/blue curves represent the negative-type/positive-type interfaces edge states. (**c**) The distribution of field for $A_{+}$, $A_{-}$, $B_{+}$, $B_{-}$ in (**b**), and the Poynting vectors are represented by red arrows.

**Figure 4 nanomaterials-12-01493-f004:**
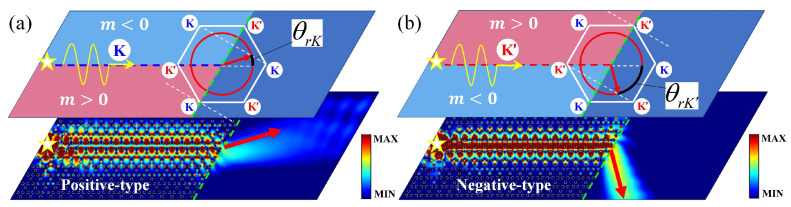
(**a**) The *k*-space analysis on out-coupling of *K* valley projected edge states along the positive-type (Zigzag) interface. (**b**) The *k*-space analysis on the out-coupling of $K^{\prime }$ valley projected edge states along the negative-type (Zigzag) interface. The white solid hexagon represents the first Brillouin zone, and the red solid circles show the dispersion in background material Si. The simulated distribution of fields at the frequency f=0.102 THz (in bandgap) are separately illustrated in the bottom panels. The light-red and light-blue regions represent m>0 and m<0, respectively.

## Data Availability

The data presented in this study are available on request from the corresponding author.

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
