# Peer review of "Topological Refraction in Kagome Split-Ring Photonic Insulators"

_nanomaterials, 2022, doi:10.3390/nano12091493_

Round 1

Reviewer 1 Report

In this paper the new kind of valley-Hall-like photonic insulator is proposed. 2D photonic insulator is the Kagome lattice formed by three split-ring resonators. The photon insulator lettice is characterized by $C_{3v}$ symmetry. It was shown that the symmetry can be broken  as by the both global rotation and the individual turns of the rings. It results in the topological phase transition. It is known that direction of outgoing beam of topological edge states depends on the type of valley from which the edge state is projected. This result was confirmed as a result of the numerical simulation carried out in this work.

The main result of the work is the proposed new structure of a photonic crystal with controllable properties of a topological reflector. 
The paper contains results that may be representing interest for experts in the photonics. I believe the
manuscript is suitable for publication.

Author Response

Thank you very much.

Reviewer 2 Report

The article on topological refraction in Kagome split-ring photonic insulators by H. Li et al. embodies an extremely interesting piece of work. The authors have done a fine job in introducing a complex topic which is of immense importance for the low-dimensional photonic community. The paper is brief and to-the-point and well organized. It covers the basic needed to understand the topic and defines valley, topological transitions and photonic band structure in Kagome lattices neatly. It also points the reader to the appropriate references in Chern number. Had it not been for the extremely poor grammar and language construction, I would have recommended publish as-is. Therefore I recommend to the authors to substantially rectify the grammar and hence make this draft a pleasure to read. With this refinement, the article is ready to be published.

Author Response

Thank you very much. We have checked the whole manuscript and corrected the grammar errors in the revised manuscript.

Reviewer 3 Report

The authors studied topological refraction in kagome lattices shaped by triple C-type unit-cell, called kagome split-ring photonic insulators. They report topological phase transition due to the C_{3v} symmetry breaking using two parameters which represent global and local rotations and valley-dependent refractions at the end of interface manipulated by different mass. The results are pretty exciting and fruitful investigations for this topic and journal. The presentation is very intuitive and logical to understand the topological origin of the phenomena employing the nontrivial mass terms caused by the band inversion. Moreover, the manuscript is well-organized and fluently written. I have simple questions for improving this article.

  • The authors used the Rayleigh scattering regime and showed the band structure in Figure 3. Could you explain why the blue band is relatively flat near Gamma point?
  • In addition, the wavelength is around 10a for the Dirac point energy in my rough estimation. You checked the robustness against the defect and impurity in appendix C. In that case, we cannot distinguish the topological robustness and ill-distinguishment of smaller-size impurity/defect than the wavelength. If you check the comparable impurity/defect with wavelength, we can agree.
  • In Appendix A, could you specify whether the surface values are eigenfrequency at the \Gamma point on the kagome lattice or others?

I hope that my questions make this manuscript improve.
